# Identification of Characteristic Points in Multivariate Physiological Signals by Sensor Fusion and Multi-Task Deep Networks

**DOI:** 10.3390/s22072684

**Published:** 2022-03-31

**Authors:** Matteo Rossi, Giulia Alessandrelli, Andra Dombrovschi, Dario Bovio, Caterina Salito, Luca Mainardi, Pietro Cerveri

**Affiliations:** 1Department of Electronics, Information and Bioengineering, Politecnico di Milano, 20133 Milan, Italy; giulia.alessandrelli@polimi.it (G.A.); andra.dombrovschi@mail.polimi.it (A.D.); luca.mainardi@polimi.it (L.M.); 2Biocubica SRL, 20154 Milan, Italy; dario.bovio@biocubica.it (D.B.); caterina.salito@biocubica.it (C.S.)

**Keywords:** sensor fusion, deep convolutional networks, multivariate signals, multitask networks, explainable AI, wearable devices, cardiovascular, signal segmentation, signal classification

## Abstract

Identification of characteristic points in physiological signals, such as the peak of the R wave in the electrocardiogram and the peak of the systolic wave of the photopletismogram, is a fundamental step for the quantification of clinical parameters, such as the pulse transit time. In this work, we presented a novel neural architecture, called eMTUnet, to automate point identification in multivariate signals acquired with a chest-worn device. The eMTUnet consists of a single deep network capable of performing three tasks simultaneously: (i) localization in time of characteristic points (labeling task), (ii) evaluation of the quality of signals (classification task); (iii) estimation of the reliability of classification (reliability task). Preliminary results in overnight monitoring showcased the ability to detect characteristic points in the four signals with a recall index of about 1.00, 0.90, 0.90, and 0.80, respectively. The accuracy of the signal quality classification was about 0.90, on average over four different classes. The average confidence of the correctly classified signals, against the misclassifications, was 0.93 vs. 0.52, proving the worthiness of the confidence index, which may better qualify the point identification. From the achieved outcomes, we point out that high-quality segmentation and classification are both ensured, which brings the use of a multi-modal framework, composed of wearable sensors and artificial intelligence, incrementally closer to clinical translation.

## 1. Introduction

Over the past few years, evidence has been accumulating in support of the view that continuous and long-term remote/home monitoring of respiratory and cardiovascular parameters is potentially feasible by leveraging a new generation of wearable sensors, capable of collecting physiological signals overnight and during daytime [1,2,3,4,5,6]. Some recent smart-watch (Heartisans^®^ (Heartisans, Hong Kong), CareUp^®^ (Farasha Labs, Paris, France), Actiwatch^®^ (CamNtech Ltd., Fenstanton, UK), AppleWatch^®^ (Apple, Cupertino, CA, United States)), chest-worn [7,8] and throat-worn [9] devices have been documented in the path of medical certification and clinical/interventional studies. On the other hand, at least three main barriers may be identified that might be slowing down the spreading of such devices in the medical domain. First, in order to gather the variety of signals, useful to describe to a large extent the physio-pathological status of the patient, the use of multiple body sensors (accelerometer, electrode, optical, acoustic, air pressure, airflow transducers) is seemingly necessary, implying obstructive and comfortless setups. Despite miniaturization and energy consumption issues, sensor integration may address this barrier [2]. Along this line, smartwatches embed heterogeneous sensors and are potentially usable for heart rate (HR), oxygen saturation (SpO2), and even blood pressure (BP) monitoring [10,11]. However, while HR monitoring, measured at the wrist, has been described as reliable [12,13], other clinically-relevant parameters suffered from uncertainty [14,15], highlighting as chest sensor measurements may achieve greater confidence. Nonetheless, the value of the combination of the information conveyed by heterogeneous sensors has been only recently investigated [7,16], its real effectiveness still remains elusive though. Second, as the measurements are collected in uncontrolled settings, motion and environment artifacts sensibly affect the signal quality complicating the extraction of accurate parameters [17]. This makes it therefore essential to associate a quality value with the computed parameters to better inform the user about the reliability of the measure [18]. Third, while the use of wearable devices allows for extensive and unique collection of physiological signals, the increasing amount of data requires an ever-increasing degree of automation in processing, highlighting caveats to traditional methods [19], and paving the way for new data-driven approaches relying upon machine and deep learning [20,21,22,23,24]. In order to address the three above issues, this paper proposes to couple a chest-worn apparatus [8], called Soundi (Biocubica Srl, Milan, Italy), able to concurrently register electrocardiogram (ECG), photopletismogram (PPG), phonocardiogram (PCG), and seismocardiogram (SCG), with a novel neural architecture, called eMTUnet (enriched multi-task Unet), to identify cardiovascular-related characteristic points. eMTUnet consists of a single all-in-one deep network able to perform three tasks simultaneously: (i) identification of fiducial points (labeling task), (ii) assessment of the quality of the signals (classification task); (iii) estimation of the classification confidence (reliability task). eMTUnet processes in a bundle all the four acquired signals (sensor-fusion), is a complete end-to-end model, avoiding any prior computation of engineered features, and finally labels in the output the quality of the processed signals. The preliminary feasibility of the proposed method was carried out by exploiting overnight recordings on healthy subjects.

### 1.1. Background

Peak detection and signal derivative analysis are state-of-the-art methods for calculating the timing of reference points in ECG and PPG. Despite ECG waveform, the PPG waveform is highly contingent on the measurement location [25] and is overall subject-dependent [1,18]. In addition, systematic noise due to motion artifacts [26,27,28] can easily distort the waveform by, for example, flattening the end of the diastolic phase, making traditional processing techniques unreliable [29]. Adaptive filtering techniques based on independent component analysis and Kalman filtering were proved unreliable in the presence of arrhythmia, as in the case of atrial fibrillation [12]. More robust PPG processing was based on sensor-fusion, also leveraging machine and deep learning methods. The use of PPG pair and inertial sensors in wrist devices increased the quality of events detection. However, the setup required the simultaneous use of both hands (the wrist of one hand and a fingertip of the other hand), limiting unrestricted movements and long-term measurements [17,30]. ECG, PPG, and pulse pressure wave sensors were used to simultaneously acquire and process signals, and ultimately estimate the blood pressure [16]. However, the setup required the use of electrodes and sensors at the wrist and ankles, limiting its use to clinical and laboratory settings. PPG and acceleration signals were processed in bundle using long short-term memory (LSTM) to improve the quality of HR estimation during physical exercise [31]. While the authors assumed that low acceleration intensity might indicate that the PPG signal is less distorted by motion artifacts, they found that the sudden changes from HR were not correctly estimated even though the acceleration intensity was low, so the actual value of such sensor integration is not clear. In order to detect artifacts in the PPG signal, 1-D CNN was proposed [23], the localization of relevant points in the waveform was not carried out though. In PCG, conventional signal processing based on the Mel-scale wavelet transform and Shannon energy has been shown to be effective only for steady-state acquisitions [19,20]. The processing of simultaneous PPG and PCG signals was exploited in [32] by integrating the PPG waveform into the calculation of the heart sound envelopes using Shannon entropy. However, the developed methodology was strictly related to and effective for the PPG waveforms detected at the finger. As documented in [15], timings and waveforms sensibly varied on the measurement site. In addition, the detection quality was verified only in a lab setting (sitting posture), and robustness to artifact motion was not investigated. Deep convolutional neural networks (CNNs) have recently been proposed as an alternative to traditional signal processing techniques for ECG, PPG, and PCG processing. Phonocardiogram was processed to compute the duration of S1 and S2 using a 1D CNN [20]. To learn explicit temporal relationships within the data, deep recurrent neural networks have been proposed to solve the task of segmenting heart sounds [21]. Simultaneous PPG and PCG signals were analyzed by a deep CNN to extract characteristic points [33]. Deep CNNs were used as well in conjunction with underlying hidden Markov models to segment PCG signal [34]. CNN and LSTM were proposed to segment the diastole, S1, systole, and S2 regions of PCG signal heart sounds [35,36]. Periodic-coded deep autoencoder networks were proposed to separate mixed heart-lung sounds under the assumption of different frequency patterns between heart rate and respiration rate. All of these papers were concerned with the labeling of individual signals of physiological episodes (S1 event, systole, S2 event, and diastole) with emphasis on the associated time window, but none of them were aimed at determining exact time instants of representative events. The seismocardiogram, representing body vibrations induced by the heart beat, is recovered by raw acceleration data, recorded at the thorax [37]. SCG contains information on cardiac mechanics, in particular heart sounds and cardiac output. A chest accelerometer was used to compute the SCG and determine a proxy of the electrocardiogram by correlation analysis [38]. Dynamic-time feature matching was used to compare the recorded SCG with a reference template with the aim of estimating the quality of the signal [39]. Based on automatic annotation, SCG was proposed as heartbeat detector, however requiring ECG for subject-specific calibration [40]. Concurrently, some deep learning approaches have been proposed to automatize the extraction of fiducial points in the seismocardiogram. A convolutional variational autoencoder network was developed to detect the heart beat [41]. ECG and SCG were concurrently measured and processed by a U-net-based cascade network to estimate robustly the respiration rate [42]. Most of these contributions focused on a single signal or signal pair, with no cases exploiting systematically sensor-fusion and signal quality.

### 1.2. Work Contributions

The present work aims at investigating the feasibility of a novel neural architecture to process in bundle four physiological signals, namely, 1-lead ECG, PCG, PPG, and SCG, simultaneously acquired with a wearable sensor, and extract fiducial points. These are fundamental to compute various clinical variables such as HR, pulse transit time (PTT), pulse arrival time (PAT), HR variability, respiration rate. Especially, PPT and PAT have been extensively described as a proxy of the BP [43,44,45]. The following main innovations outline the work scope:the eMTUnet architecture evolves the traditional regression networks by a multi-task model that enriches the identification of characteristic points with the prediction of the quality of each signal, along with an overall quality of the signal bundle;exploiting the simultaneous acquisition of the four signals, feature-to-feature fusion in the eMTUnet unites the cues from all of the four sensors enabling data redundancy and likely mitigating the effect of artifacts. Especially, R-peak in ECG, S1 sound in PCG and maximum slope point in SCG can be considered to convey a similar information (onset of the systolic wave);the eMTUnet model exploits a novel confidence score, based on Jain fairness index, to qualify the signal classification, which might be used to inform to end-user about the reliability of the identified points.

## 2. Materials and Methods

### 2.1. Wearable Device

Every signal involved in this study was acquired with the same non-invasive device, named Soundi (Biocubica Srl, Milan, Italy). This apparatus is positioned on the chest of the subject close to the heart and features a simultaneous recording of reflective photoplethysmographic, phonocardiographic, electrocardiographic, and accelerometer signals. The device has been recently patented (Patent No. EP3248541A1) at a European scale and is currently under CE marking procedure as a medical device (class II). The current release (Figure 1) has a circular shape with a diameter of about 6 cm and a thickness of about 1 cm, and it weighs no more than 40 g. In order to minimize the invasiveness of the device, it can be attached to the thorax using double-sided adhesive medical-certified tape, making the removal phase very easy. Technically, it is a fully integrated device with its power supply, microprocessor, acquisition sensors, data storage (removable micro SD), and wireless communication. Three electric terminals are devoted to plugging wired electrodes placed on the thorax to acquire a single derivation ECG analog signal (1-lead ECG recorder), which is digitized directly on board, usable primarily for basic heart monitoring and bioimpedance. PPG signal is measured through a digital optical sensor. The acoustic signal is recorded using a digital air pressure sensor coupled with a custom-designed bell-shaped chassis (details about principles, design and implementation can be found in [8]). One digital six-axis inertial measurement unit (IMU) records the acceleration and angular speed encoded in a quaternion, usable for seismocardiogram computation, posture and motion detection. The main functionalities (e.g., start and stop time, acquisition protocol and duration, battery status, enabling/disabling sensors, signal quality verification) are controllable via a smartphone-based application connected to the apparatus by low energy Bluetooth (BLE) communication. The electrical power is supplied by an internal lithium-ion battery, being rechargeable directly using a USB-C cable. While each digital sensor has its native default sampling rate, the acquisition frequency can be adapted according to specific requirements and set up to the kHz range for PPG, PCG, and ECG signals. At the end of the acquisition, the micro SD can be easily removed from the device and coupled to a PC for data download and processing. Further technical details and performances were earlier reported in [8].

### 2.2. Data Collection Protocol

Five male volunteering participants who had no previous signs of cardiovascular diseases were recruited for the feasibility study. On average, they were 42.5 years old (25–60). Subject enrollment and data acquisition were performed according to the descriptive rules of the experimental protocol (Opinion 3/2019, dated 19 February 2019) that received approval by Politecnico di Milano Ethical Committee. All participants were provided with all the required information about the experimental sessions, and they were asked to sign an informed consent prior to the tests. The participant was asked to wear the Soundi sensor, attached with medical tape on chest surface in correspondence of the left second intercostal space. The three ECG electrodes were positioned on the left and right clavicle, and lower left abdomen. Data acquisition consisted of a single overnight recording (about 8 h) at home. All four signals were sampled at a frequency of 400 Hz.

### 2.3. Data Pre-Processing

Acquired ECG, PCG, PPG, and accelerometer (ACC) signals were transferred to a PC and underwent pre-processing by a custom SW application developed in Python. All the signals were filtered through a Butterworth bandpass filter (details in Table 1) to remove low-frequency components caused by breathing, as well as high-frequency components caused by muscle vibrations and acoustic noise. Acceleration signal was properly filtered to keep cardiac vibrations unaltered. In order to obtain the seismocardiogram, the filtered accelerometer signal was firstly reduced to a single magnitude component, called AM as:(1)$$AM=\sqrt{ACC{{}_{x}}^{2}+ACC{{}_{y}}^{2}+ACC{{}_{z}}^{2}}$$
where $ACC_{x}$, $ACC_{y}$ and $ACC_{z}$ are the three orthogonal acceleration components. Then, Hilbert envelop and Savitzky-Golay filtering (151 samples and four coefficients) were sequentially applied to AM to compute the SCG [46]. This processing synthesizes the SCG into two main patterns representing the aortic valve opening and closure respectively (Figure 2).

### 2.4. Signal Annotation

Out of the 8 h of acquisition, about 2.5 h (in the central part of the night) were selected for the annotation in each recording, leading to twelve and a half hours overall. Due to the multi-task nature of the framework, the annotation of the data included both a segmentation and a classification phase. Both types of annotations were done semi-automatically through a custom interactive dashboard (developed in Python exploiting Voilà platform) that allowed the signals to be visually inspected and labeled interactively. Segmentation aimed to find specific landmarks in each signal (Figure 3) as:ECG: peak of the R wave in each heart beat;PCG: center of S1 and S2 sound envelopes;PPG: point featuring the maximum slope in the systolic wave;SCG: point featuring the maximum slope in the rising vibration signal preceding the aortic valve opening.

Regardless of the signal, each landmark was first identified manually. Its position was then refined based on the closest point that matches the selected criteria for that signal (e.g., the maximum of the signal, the maximum slope, etc.). A binary signal was generated where the value one and zero identified the occurrence and the absence of the landmark, respectively. A 10 ms time window was finally used to embed the landmark so that, considering the sampling frequency of 400 Hz, the annotated landmark was encoded by 4 samples in the binary signal.

The classification of the signal quality (Figure 3), performed by visual inspection using a sliding window, was based on four main types of trends as follows:class 1: all signals had an overall good quality; landmarks were easily detectable.class 2: PCG was corrupted; the other signals were basically good, and landmarks were pretty detectable.class 3: PPG was corrupted; the other signals were basically good, and landmarks were pretty detectable.class 4: motion artifact; overall the four signals was mostly corrupted, landmarks were barely detectable.

The final step in creating the training dataset was to divide the segmented and classified sample collection into chunks of three different lengths, namely, 2, 4, and 6 s, respectively. If a chunk contained samples belonging to more than one class, that chunk was classified as belonging entirely to the worst class. Three datasets, called $D_{2}$, $D_{4}$ and $D_{6}$, were built by collecting chunks of the same length (2, 4 and 6 s, respectively). Given a sampling frequency of 400 Hz, one chunk for $D_{2}$, $D_{4}$ and $D_{6}$ corresponded to a signal of 800×4, 1600×4 and 2400×4 samples, respectively. In addition, 40% of the data were overlapping chunks, resulting in 42,746 for $D_{2}$, 21,370 for $D_{4}$ and 14,244 chunks for $D_{6}$ (Table 2). Interestingly, it was found that the amount of class 3 chunks (corrupted PPG) was more than twice the amount of class 2 chunks (corrupted PCG), while a percentage less than 10% of chunks was in class 4. As the last step, every chunk for each of the four signals (ECG, PCG, PPG, SCG) was standardized in order to have mean zero and variance equal to one. Overall, the data annotation, performed by A.D and D.B., took about 80 h.

### 2.5. Deep Convolutional Neural Network Architecture

The developed network, named enriched multi-task Unet (eMTUnet), processed a four-stacked signal tensor to produce a corresponding labeled signal bundle in the output, along with a classification for the quality of the bundle and a numerical measure of the classification quality overall. The architecture proposed in this work is adapted starting from the U-shaped network model, a neural network that was extensively tested in image segmentation [47]. It is based on an encoding-decoding structure with horizontal skip connections between the encoder and decoder paths. With respect to traditional Unet, eMTUnet exploits a squeeze-and-excitation (SE) block after every stage of encoding and decoding, to enhance informative feature maps while suppressing less helpful ones [48]. The SE block (Figure 4) comprised a global average pooling layer that compressed each feature map, resulting in a single value per channel. This layer was followed by a fully connected layer featuring a ReLU activation function that compressed the number of channel by halving it. Then, another fully connected layer with a sigmoidal activation function restored the initial number of channel. Finally, each input feature map was weighted by multiplication with the last convolutional layer. The SE blocks added a content-aware mechanism to weigh each channel adaptively. To summarize, the excitation operator assigned a set of channel weights to the input. In this sense, SE blocks naturally incorporated input-conditioned dynamics, which may be seen as a self-attention function on channels whose connections are not limited to the local receptive field to which the convolutional filters are sensitive.

With respect to traditional Unet architecture, the basic convolutional block were substituted by inception blocks (Figure 5). This block was adapted from GoogLeNet [49] and its main purpose was to perform multi-scale feature extraction from the input signals. Each stage was made by a 1-D convolutional layer with kernel size 3×1, 7×1, and 11×1 followed by batch normalization. The input received by this block was processed simultaneously by multiple receptive fields, and the output of each branch was then concatenated. The sequence of convolution and batch normalization layers was repeated twice per stage. Batch normalization was employed for better generalization and faster training.

The overall eMTUnet architecture (Figure 6) was characterized by three layers in the encoder block and three in the decoder one. Blocks in the decoding path were connected by a max-pooling layer that compressed, by halving, the signal size while doubling the number of feature maps. The decoding path performed the same operation but with an upsampling layer substituting the max-pooling, doubling the size while halving the feature maps. All convolutional layers in both encoding and decoding paths exploited the Leaky Relu activation function, with an α parameter equal to 0.1. Convolutional layers with one feature map and a kernel size of one with a sigmoid activation function were used to generate segmentation results. On the other hand, the classification output was generated through a convolutional layer with five feature maps, a kernel size of one, and the Softmax activation function. The final probability of class membership result was then achieved by compressing the convolutional layer with five feature maps through a global averaging. The last part of the network involved the generation of an estimate of the classification confidence. It consisted of two fully-connected layers with four and one neurons, respectively, featuring a sigmoidal activation function neuron. This last block was trained separately from the rest of the network, as better clarified in Section 2.6.

### 2.6. Reliability of the Prediction

Classification output was presented as a 4-elements array containing the expected probability that a sample belonged one of the four classes described in Section 2.4. The maximum probability among these four values designated the actual output class. If the difference between the highest probability and the second one was significant, the prediction could be considered very confident. On the other hand, if this difference was slight, confidence in classification dramatically decreased accordingly. In order to quantitatively evaluate the network classification confidence, Jain’s fairness index (JF) [50] was computed for each sample in the training dataset as:(2)$$JF\left(y_{1},y_{2}\right)=\frac{{\left(\sum_{i=1}^{2}y_{i}\right)}^{2}}{2\cdot \sum_{i=1}^{2}y_{i}^{2}}$$
where $y_{1}$ is the maximum value in the classification prediction array and $y_{2}$ is the second one. A high fairness score indicated an uncertain model prediction as the two-top classes were predicted with very similar probabilities. On the other hand, a low JF value ensured very high confidence in the class predictions. As, the score was bounded in the [0.5,1] range, a min-max normalization was applied to remap it in [0,1] range, considering the entire training dataset, obtaining the normalized fairness score $\hat{JF}$. Finally, a Confidence Score (CS) was computed as:(3)$$CS=1-\hat{JF}$$

The higher the value of CS, the higher the confidence of the prediction, and vice-versa. CS values were considered as the target values for the training of the confidence output of MTUnet (cfr. Figure 6 right-most fully-connected blocks). As described above, this required to introduce the confidence neural module, consisting of two fully-connected layers in cascade featuring 25 free parameters. This network was trained (see next paragraph) separately and added to the pre-trained eMTUnet afterwards. The confidence neural module made the overall network self-contained to accomplish not only the segmentation and classification tasks but also able to quantify the confidence of the predictions, paving the way towards enabling intrinsic explainability in deep networks [51].

### 2.7. Loss Functions and Metrics

Due to the multi-task nature of the network, a different loss function was computed for each output, namely, the segmentation, classification, and confidence. Due to the intrinsic nature of the encoding of the landmark occurrence (cfr. Section 2.4), large imbalance in the label dataset was attained. Under this condition, Dice similarity loss ($DSC_{Loss}$) was documented effective for signal segmentation [52]. Likewise, the use of the F1-score as a loss function ($F1_{Loss}$) was proven effective in the simultaneous optimization of label prediction and label counts in a single task by combining precision and recall [53], as the present segmentation task needs. Thus, the implemented $Loss_{S}$ exploited a combination of the two factors as:(4)$$DSC_{Loss}\left(y,\hat{y}\right)=\frac{1}{N}\sum_{i=1}^{N}1-\frac{2y_{i}\hat{y_{i}}}{y_{i}+\hat{y_{i}}}$$
(5)$$F1_{Loss}\left(y,\hat{y}\right)=1-\left(2\cdot \frac{precision\left(y,\hat{y}\right)\cdot recall\left(y,\hat{y}\right)}{precision\left(y,\hat{y}\right)+recall\left(y,\hat{y}\right)}\right)$$
(6)$$Loss_{S}=\left(\frac{DSC_{Loss}+F1_{Loss}}{2}\right)$$
where *y* is the segmentation target mask created during the labeling phase and $\hat{y}$ is the network prediction. The classification loss ($Loss_{Cl}$), on the other hand, was based on categorical cross-entropy (CCE) computed over the four classes:(7)$$Loss_{Cl}=CCE\left(y,\hat{y}\right)=-\frac{1}{N}\sum_{i=1}^{N}\sum_{i=1}^{4}y_{i}log\left({\hat{y}}_{i}\right)$$
where *y* is the classification target class defined in the labeling phase and $\hat{y}$ is the predicted class. The classification confidence loss ($Loss_{Co}$) was based on the mean absolute error (MAE):(8)$$Loss_{Co}=MAE\left(y,\hat{y}\right)=\frac{1}{N}\sum_{i=0}^{N}\left|y_{i}-\hat{y_{i}}\right|$$
where *y* is the CS value computed after the separate training of the network classification output, and $\hat{y}$ is the predicted CS score. As far as evaluation metrics is concerned, recall, precision and accuracy were used to quantify the performance of both segmentation and overall classification tasks. In order to highlight the classification performance across the 4 specific classes, class specific receiver operating characteristic (ROC) curves, along with the corresponding area under the curve (AUC), were computed and reported. In order to quantify the confidence of the overall classification, the distribution of correctly classified examples was compared against the same distribution for misclassified examples.

### 2.8. Implementation Details

Every dataset was split into train/validation/test set with a ratio of 70%, 15%, 15%, respectively. The network models, loss functions, metrics, and training algorithms were developed in Python exploiting TensorFlow library. The training took place in a Google Colaboratory Cuda-enabled environment with a four-core CPU, 25 GB RAM, and an NVIDIA^®^ Tesla^®^ P100 GPU with 16 GB RAM support. The training was optimized using ADAM (Adaptive Moment Estimation) optimizer, with a learning rate equal to $10^{-3}$. The batch size was set to 20 samples while a total number of 500 epochs were considered. When the segmentation loss score computed on the validation set was maximized, the training routine was configured to save the set of the best network weights. An early stopping callback was set to interrupt the training whether the same loss function did not improve in the last ten epochs. On average, the training took about 25, 20 and 15 min, for $D_{2}$, $D_{4}$ and $D_{6}$ datasets respectively. The predictions of one chuck, irrespective of its time length, was practically instantaneous. The part of the network devoted to computing the class confidence output was trained separately from the rest, maintaining the same hyperparameters. Then, it was combined to the rest of MTUnet in order to create a single network capable of performing three tasks at a time, namely: (i) the identification of fiducial points inside the input signal bundle, (ii) the quality classification of the bundle itself, and (iii) an estimate of the confidence of the quality classification.

## 3. Experimental Results

The effect of SE and inception blocks in the eMTUnet network was tested against the network MTUnet, where such blocks were disabled. For the sake of clarity, in MTUnet the multi-resolution convolutional block was collapsed to a single path with a kernel size of 3 × 1. Ablation experiments were carried out across $D_{2}$, $D_{4}$, and $D_{6}$ sets comparing landmark identification errors and signal classification. According to the nature of the output encoding, the assessment of the landmark identification was draw up as a binary classification problem. Coherently, the segmentation predictions, provided by eMTUnet (four arrays of sigmoidal neurons), underwent dichotomization using a threshold of 0.5. According to the labeling, a 10 ms time-window was used to embed each obtained burst, indicating the presence of a landmark. Predicted 10 ms windowed bursts were considered true positives whether featuring non-null intersection with one existing labeled characteristic point, false positive otherwise. Whenever a label did not have a corresponding prediction, the false negative amount was incremented. For each configuration, a ten-fold record-wise cross validation was applied. Statistical analysis were performed using non-parametric Kruskal-Wallis test with a level *p* of the 5% for significant difference. Likewise, the effectiveness of the segmentation of signals in bundle (sensor-fusion) was compared to the single-signal processing using a traditional Unet without the classification task. Recall and precision of fiducial point detection in PPG and PCG were computed on the test set.

### 3.1. Labeling Task

#### 3.1.1. Effect of SE and Inception Blocks

Recall and precision results on the 10-fold cross validation test for $D_{2}$
$D_{4}$ and $D_{6}$ sets were accurate and comparable in between. As expected, the identification of the peak in the R-wave of the ECG was optimal with no false positives or false negatives. The identification of S1 and S2 sounds showed a recall greater than precision in both $D_{4}$ and $D_{6}$ sets and for both networks. The eMTUnet was statistically better than MTUnet in both recall (*p* = 0.02, *p* = 0.009, *p* = 0.0001) and precision (*p* = 0.002, *p* = 0.003, *p* = 0.002), in $D_{2}$, $D_{4}$ and $D_{6}$ sets, respectively. Interestingly, the detection of the maximum slope in the PPG was more precise than sensible. No statistical difference was found for precision (*p* = 0.13) between the two networks in the $D_{4}$ set, while eMTUnet outperformed MTUnet (*p* = 0.02, *p* = 0.007) in the $D_{2}$ and $D_{6}$ sets. eMTUnet was statistically better than MTUnet for PPG recall in all the three sets, $D_{2}$ (*p* = 0.01), $D_{4}$ (*p* = 0.006) and $D_{6}$ (*p* = 0.0002) sets. In addition, the detection in SCG showcased a greater precision that recall. However, no statistical difference was found (*p* > 0.5) either between the three sets or between the two networks.

#### 3.1.2. Role of Sensor-Fusion

Without loss of generality, the role of sensor-fusion was tested on $D_{6}$ dataset. The segmentation quality for PCG and PPG, obtained with the eMTUnet, was compared with the quality obtained by a traditional Unet, with or without the used of SE and inception blocks (Table 3). Given that eMTUnet outperformed (PCG: 0.90 and 0.74; PPG: 0.89 and 0.97) the traditional Unet, irrespective of whether SE and inception blocks were included, allowed to assert the effectiveness of processing the four signals in bundle.

### 3.2. Classification Task

Classification results, reported as the average value over the four classes, were in the range of 0.90 and 0.85 for eMTUnet and MTUnet, respectively (Table 4). The performance of the eMTUnet featured a statistical difference (*p* = 0.01, *p* = 0.002, *p* = 0.00002) in comparison to the MTUnet, again confirming the view that SE and inception blocks altogether were effective. The robustness of the eMTUnet network to the chunk length variation was further supported on whether no statistical difference (*p* > 0.5) was found across the three sets. The analysis of the ROC curves, verifying the corresponding AUC, shed light on the results over the four classes (Figure 7). Overall, class 4 was the less accurate in both tests with AUC values equal to 0.88 and 0.79, respectively. For the remaining three classes, the AUC values were all greater than or equal to 0.93. The confusion matrices showcased as well the classification accuracy of the first three classes with respect to the last one (Figure 7). This difference could be motivated as class 4 was under-represented (less than 10%) in all the three training sets (cfr. Table 2).

### 3.3. Reliability Task

The distribution of the two highest value difference in the classification output vector (cfr. Figure 8a) showed that the distribution of correctly classified examples was correctly skewed towards 100%, with median (interquartile range, IQR) value equal to 0.93 (0.16). On the other hand, the distribution of misclassified examples was pretty uniformly distributed in the [0 1] range, featuring a median (IQR) value of 0.52 (0.53). This trend reflected in the distribution of the predicted CS scores (cfr. Figure 8b), showing that correctly classified samples corresponded to higher CS scores compared to the misclassified ones. Median (IQR) values were equal to 0.94 (0.13) and 0.84 (0.49) for correctly classified and misclassified examples, respectively. As a final example, in (Figure 9) the predicted landmarks in a 6 s signal bundle, which was classified in class 1 with a CS score equal to 0.97.

## 4. Discussion

### 4.1. Main Findings

The proposed multi-task deep neural network, called eMTUnet, was applied to identify fiducial points in a bundle of 4 signals acquired simultaneously by a chest-worn device. The acquisition protocol envisaged overnight recordings into an uncontrolled home setting. The network was devised to predict concurrently the signal quality split in four different classes (cfr. Section 2.4). In order to abridge the overall prediction quality, a confidence score was enabled as an additional output of the network. From the framework point of view, the eMTUnet took its root from the traditional Unet model, commonly used for 2D or 3D image segmentation, and was evolved, in this work, into a multi-task network by joining the signal annotation task to one classification task and one prediction confidence task. With respect to the classical encoder-decoder Unet, the implemented architecture was further revised, introducing the squeeze and excitation layer in skip and decoding connections and a multi-resolution convolutional layer, shaped as an inception module. Segmentation and classification results endorsed both the advantage of SE and inception blocks, and the effective role of processing the signals in bundle, supporting the advantage of the eMTUnet against the classical Unet model. Especially, the sensor fusion approach adopted in this work comes from the consideration that the four signals can be envisioned to potentially be physiologically linked and recognized to convey some interrelated information. According to this view, signals can be assumed partially interchangeable, especially when affected by noise, enabling the sorting from high to low-quality signals. Especially, since the transducers in the ’Soundi’ device exploit different physical principles, it is reasonable to expect that noise artifacts occurring at certain times during long-term acquisition sessions will affect signals at different levels while the rest might remain unaffected. For example, during overnight acquisitions, snoring can disrupt PCG and acceleration waveforms while the PPG and ECG are not substantially affected by this disturbance. While the overall relation among the characteristic points of interest remained elusive, the results confirmed (cfr. Table 3) the advantage of using processing in bundle. In this regard, the detection of the R-peak was shown to be less critical but could be considered as a proxy for other fiducials, especially for the S1 sound in the PCG. Likewise, characteristic points in the actigram may be less relevant from a physiological point of view, however the classification of this signal may be useful to explicitly identify motion artifacts that may deteriorate the quality of the overall signal bundle. According to the temporal localization of the characteristic points (cfr. Table 5), at least 90% of the detected R-peak events in the ECG, S1 and S2 events in the PCG, and maximum systolic slope events in the PPG were found with an uncertainty of 10 ms. Assuming an average value of 200 ms for the PTT and PAT in a regular heart-beat regimen (60–70 bpm), the determination of the event occurrence with such temporal resolution, led to an error in the range of 5%, which is tolerable for computing such quantities. It can be argued therefore that the developed network can be further tested to calculate relevant cardiovascular variables. As far as network training is concerned, it is important to remind that in the annotation stage (cfr. Section 2.4) characteristic points into a signal were not labeled, when corrupted (cfr. membership in classes #2, #3, and #4), so that potential confusing data were avoided and bias reduced. By construction in the extrapolation, the network was able however to perform segmentation in all the four classes, predicting independently the most likely classification and the corresponding confidence score (cfr. Section 2.6), this last enabling the operator whether to use or not the predicted characteristic points. This facility can be regarded as an attempt to endow a deep network with an intrinsic tool to better explain/justify the prediction, composed by the regression and the classification tasks. Technically, the confidence score, based on the Jain’s fairness index, was a direct output of the network. As such, a devoted neural block was opportunely trained offline, with respect to the segmentation and classification task, and then linked to the eMTUnet.

### 4.2. Comparison with the Literature

Pre-trained CNN models (AlexNet, VGG16, and VGG19) were applied to the processing of acoustic signals, registered with digital stethoscopes, to discriminate normal S1 and S2 sounds from murmurs and artifacts [54]. Precision results for both normality and abnormality were not greater than about 80%, being similar to more traditional detection methods [32]. In addition, the recordings were performed in lab-controlled settings, which strongly reduced artifacts. In order to segment physiological episodes (S1 event, systole period, S2 event, and diastole period) in the PCG signal, CNN model with long-short-term memory (LSTM) layers were proposed [35,36]. However, the model was designed to detect the time period embedding the episode, disregarding again the exact location of the correspondent fiducial points. A 1D CNN model was trained to classify 5 s PPG segments, recorded on 30 subjects, into two classes (clean or artifact-affected), achieving about 95% of accuracy on the test dataset (8 subjects) [23]. Despite the obtained high accuracy, the work results had a reduced span, being the deep network limited to a binary classification disregarding the focus on characteristic points. LSTM networks were proposed to detect heart abnormality exploiting heart sound identified in the PCG [55]. The classification results of about 80% (AUC scores) were higher than those obtain by applying traditional phonocardiogram processing methods based on Mel Frequency Cepstral Coefficients. Again, the detection only featured a binary classification. Along the same path, supervised machine learning techniques were proposed to assess the quality of the 30 s chuck PPG, comparing fingertip recording and wearable commercial wristband [18]. Results did not highlight differences in the two acquisition modalities, featuring a discrimination accuracy between pathological and poor quality signals of about 95%. However, signal were collected under controlled conditions in intensive care unit avoiding exogenous artifacts. Overall, these works did not explicitly face issues about the confidence of the predictions. In this regard, gradient-weighted class activation mapping (Grad-Cam) was exploited in a convolutional residual deep neural network to process 12-lead ECG multi-array signal and visualize the most important regions in the signals accounting the most for multi-label classification task where free-text physician annotations were included as additional input [56]. However, the data acquisition was performed in ambulatory settings and limited to the ECG only. Likewise, ECG was processed throughout neural networks providing descriptive scores for describing P-wave irregularity and possible atrial fibrillation [57]. Despite the interesting attempt to deal with explainability in the deep networks, the work span was limited to a single signal.

### 4.3. Technical Challenges and Work Limitations

The generalization quality of deep learning models depends greatly upon on the labeling (data amount and precision) performed to build the training set. In this work, labeling was performed manually and carefully revised, which took an intensive effort. Annotation of signal quality into the four classes was basically subjective. No quantitative evaluation of the signal to noise ratio was performed. The study cohort was limited to 5 subjects, however, more than 12 h of signals were annotated in total and used to training and test the developed model. In addition, overnight recording, in a home uncontrolled settings, ensured a large signal variability, potentially introducing noise because of motion artifacts due to snoring and change of sleeping postures. Additional variability of the signal quality was detected before the true sleeping phase and because of awaking in the early morning. It is well documented actually that polysomnography in general is especially prone to the so-called ‘first-night effect’, in which the first night of experiment shows more sleep fragmentation, longer initial sleep delay, less total sleep time, and more wakefulness when compared to subsequent nights [58]. In our specific case, we decided a posteriori to retain the central part of the night (about 2.5 h) that showed enough signal stability for all the 5 subjects. Despite this choice, a certain variety in the signal quality was attained (see Table 2 for the distribution in the four quality classes). Given the specificity of the ‘Soundi’ device that recorded ECG, PCG, PPG, and SCG, we developed a model to process them together. However, we could not test our model with public data such as Physionet (https://physionet.org, accessed on 1 February 2022) because there was no availability of simultaneous acquisitions.

## 5. Conclusions

The proposed method applies multi-task deep learning to event labeling (segmentation and classification) in multivariate physiological signals, as an alternative to labor-intensive manual detection. From the achieved outcomes, we point out that high-quality segmentation and classification are both ensured, which brings the use of a multi-modal framework, composed of wearable sensor and artificial intelligence, incrementally closer to clinical translation. As the sample size is small for this preliminary feasibility study, future work includes planning and implementing a new acquisition protocol applied to monitoring the sleep in a cross-sectional study.

## Figures and Tables

**Figure 1 sensors-22-02684-f001:**
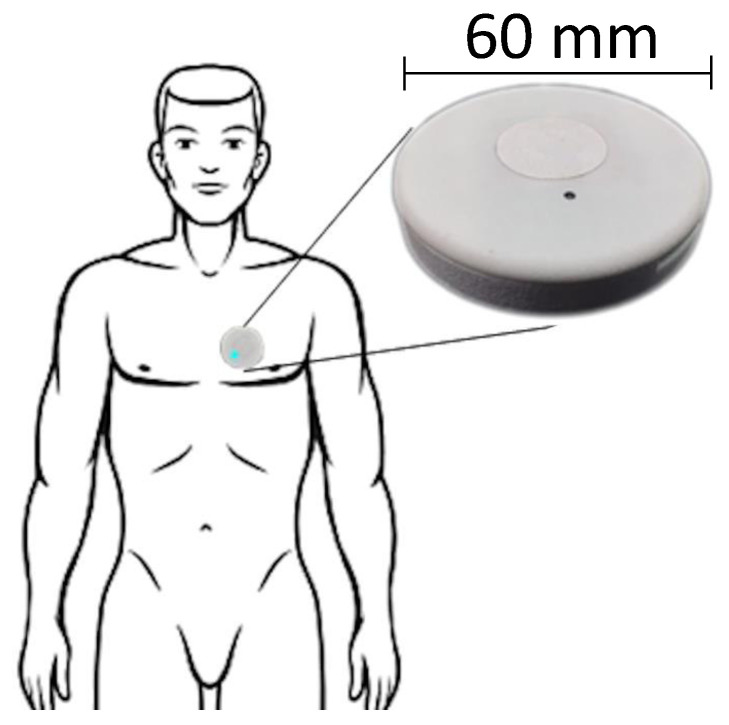
Soundi wearable device.

**Figure 2 sensors-22-02684-f002:**
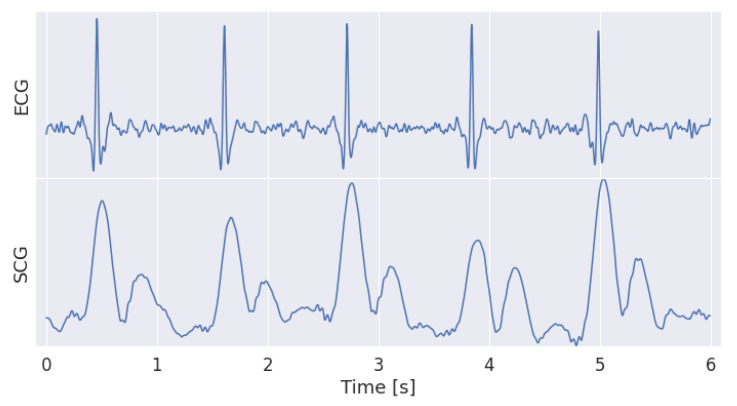
Comparison of ECG and SCG signals. As expected, the envelop of the acceleration signal highlights the two patterns, the opening and the closure of the aortic valve, in the SCG waveform. Coherently, the R wave in the ECG slightly anticipate the opening of the aortic valve.

**Figure 3 sensors-22-02684-f003:**
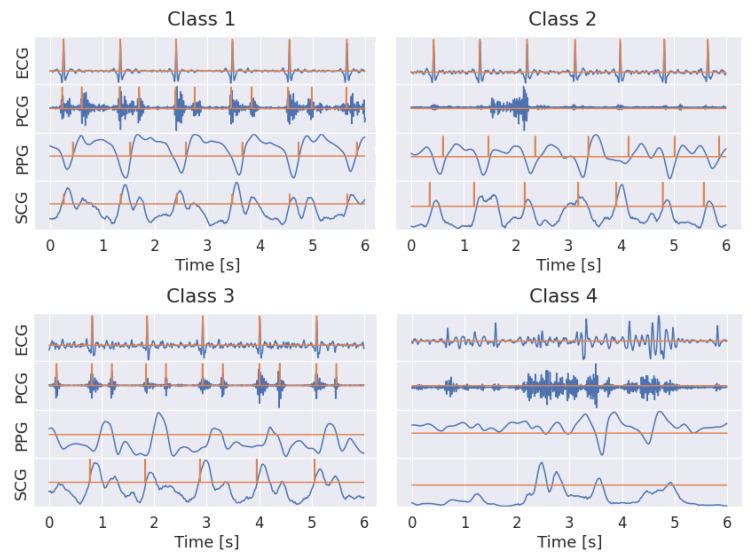
Examples of labeled signals in the four different classes. ECG, PCG, PPG and SCG are represented in row from top to bottom. The label is represented by a binary signal, featuring 1 in correspondence of the detected characteristic point. In class 1, the typical SCG pattern, featuring two peaks corresponding to cardiac sounds, can be noticed.

**Figure 4 sensors-22-02684-f004:**
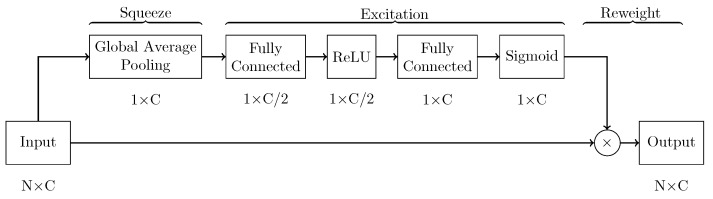
Squeeze and excitation (SE) block.

**Figure 5 sensors-22-02684-f005:**
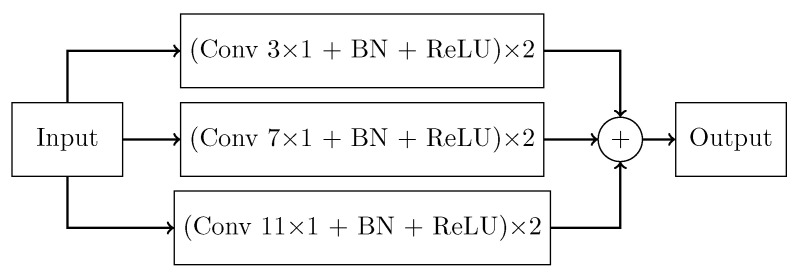
Inception block.

**Figure 6 sensors-22-02684-f006:**
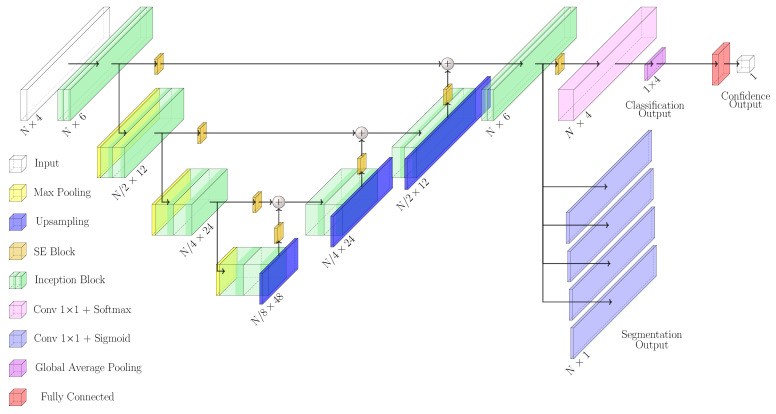
eMTUnet architecture. Some 70 thousand free parameters defined the network.

**Figure 7 sensors-22-02684-f007:**
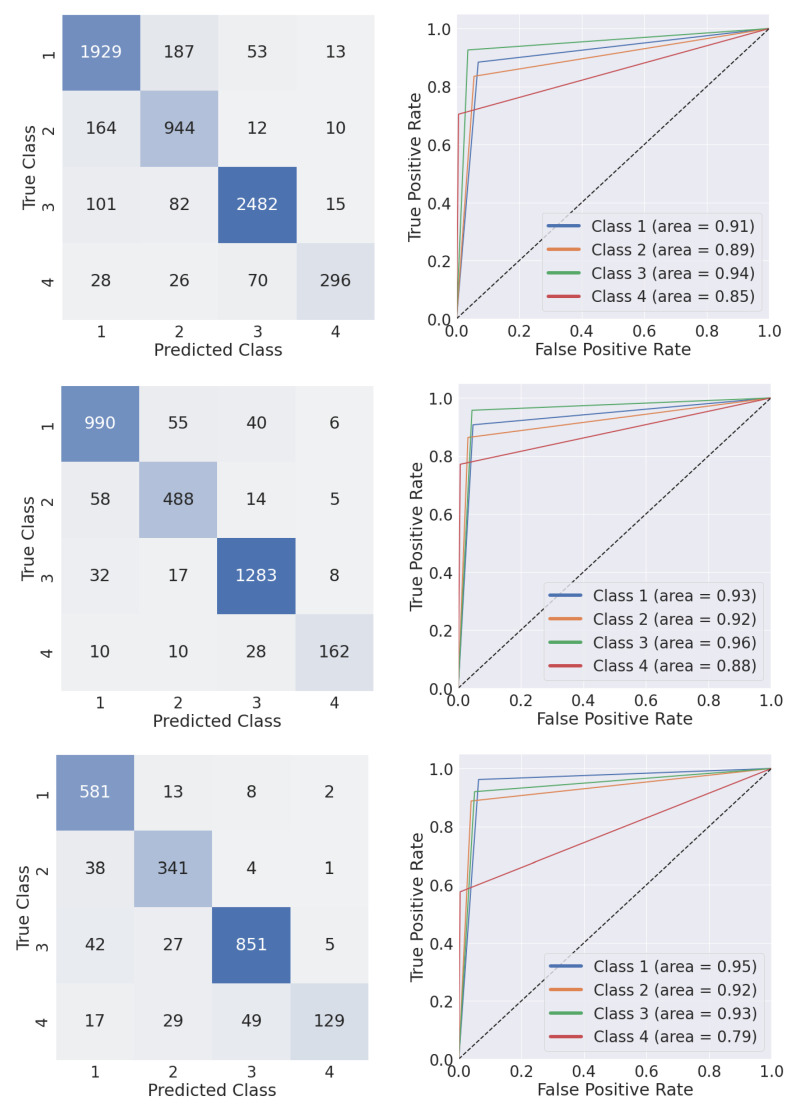
Multi-class confusion matrix and class-specific receiver operating characteristic curves computed with eMTUNet model trained with $D_{2}$ (**top** row), $D_{4}$ (**central** row), and $D_{6}$ (**bottom** row) test set.

**Figure 8 sensors-22-02684-f008:**
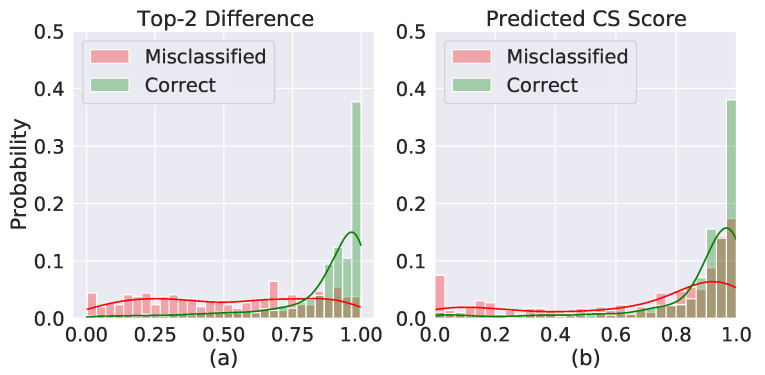
(**a**) Distribution of differences between the two highest values in the classification output vector for correct classified samples (green bars) and misclassified ones (red bars). (**b**) Distribution of predicted CS scores for correct classified samples (green bars) and misclassified ones (red bars). Both distributions were computed with $D_{6}$ test set.

**Figure 9 sensors-22-02684-f009:**
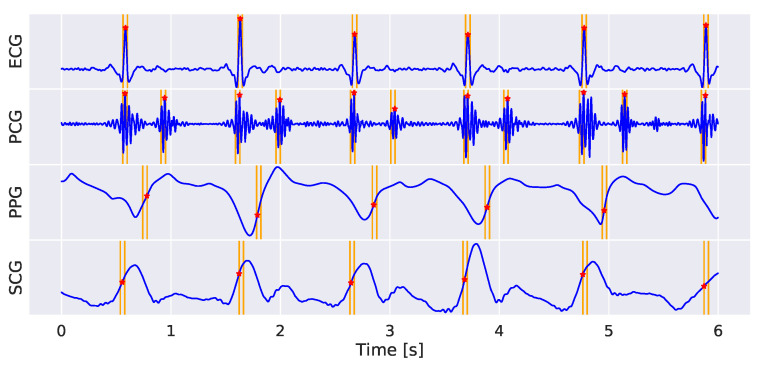
Example of a 6 s acquisition with superimposed eMTUnet predicted segmentation landmarks. The network classified this example as class 1 with a CS score equal to 0.97.

**Table 1 sensors-22-02684-t001:** Bandpass filter parameters applied to the signal of interest. The accelerometer (ACC) parameters were applied separately to all the three components (*x*, *y*, and *z*).

Signal	Order	Bandpass Frequency [Hz]
ECG	3	[5, 40]
PCG	4	[25, 50]
PPG	2	[1, 4]
ACC	4	[5, 15]

**Table 2 sensors-22-02684-t002:** Amount of chunks, along with the corresponding percentage out of the overall number, labeled in the four classes for all the three training datasets.

Class	D${}_{2}$		D${}_{4}$		D${}_{6}$	
1	14,282	33%	7199	34%	4686	33%
2	7603	18%	3786	18%	2552	18%
3	18,095	42%	8978	42%	6032	42%
4	2766	6%	1407	7%	974	7%
	42,746		21,370		14,244	

**Table 3 sensors-22-02684-t003:** Comparison of the segmentation quality, tested on $D_{6}$ dataset, between the proposed eMTUnet and the traditional Unet (single input signal and single task), with or without SE and inception (I) blocks.

Model	PCG		PPG	
	Rec	Prec	Rec	Prec
Unet	0.87	0.64	0.66	0.84
Unet + SE + I	0.89	0.62	0.55	0.79
eMTUnet	0.90	0.74	0.89	0.97

**Table 4 sensors-22-02684-t004:** Comparison of metrics performances for eMTUnet and MTUnet. Accuracy, recall, and precision are computed for classification task. Each value was evaluated with a ten-fold cross-validation on both $D_{2}$, $D_{4}$, and $D_{6}$.

Dataset	Model	Accuracy	Recall	Precision
$D_{2}$	MTUnet	0.85 (0.02)	0.75 (0.07)	0.85 (0.07)
	eMTUnet	0.89 (0.01)	0.84 (0.02)	0.87 (0.01)
$D_{4}$	MTUnet	0.86 (0.01)	0.77 (0.03)	0.87 (0.01)
	eMTUnet	0.91 (0.01)	0.86 (0.01)	0.90 (0.01)
$D_{6}$	MTUnet	0.86 (0.02)	0.76 (0.04)	0.86 (0.01)
	eMTUnet	0.91 (0.01)	0.88 (0.02)	0.89 (0.02)

**Table 5 sensors-22-02684-t005:** Comparison of metrics performances for MTUnet and eMTUnet. Recall and precision are computed for landmark identification task. Each value was evaluated with a ten-fold cross-validation on both $D_{2}$, $D_{4}$, and $D_{6}$.

Dataset	Model	ECG		PCG		PPG		SCG	
		Rec	Prec	Rec	Prec	Rec	Prec	Rec	Prec
$D_{2}$	MTUnet	1.00 (0.00)	1.00 (0.00)	0.87 (0.02)	0.73 (0.01)	0.82 (0.01)	0.95 (0.01)	0.81 (0.00)	0.88 (0.02)
	eMTUnet	1.00 (0.00)	1.00 (0.00)	0.91 (0.01)	0.79 (0.03)	0.89 (0.02)	0.98 (0.01)	0.81 (0.02)	0.88 (0.02)
$D_{4}$	MTUnet	1.00 (0.00)	1.00 (0.00)	0.86 (0.02)	0.72 (0.02)	0.81 (0.02)	0.96 (0.02)	0.79 (0.03)	0.87 (0.03)
	eMTUnet	1.00 (0.00)	1.00 (0.00)	0.90 (0.02)	0.77 (0.01)	0.88 (0.03)	0.97 (0.02)	0.82 (0.04)	0.89 (0.02)
$D_{6}$	MTUnet	1.00 (0.00)	1.00 (0.00)	0.85 (0.02)	0.70 (0.02)	0.81 (0.05)	0.95 (0.02)	0.80 (0.03)	0.87 (0.03)
	eMTUnet	1.00 (0.00)	1.00 (0.00)	0.90 (0.01)	0.74 (0.03)	0.89 (0.03)	0.97 (0.01)	0.81 (0.02)	0.87 (0.03)

## Data Availability

The data presented in this study are available on request from the corresponding author.

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
