# Peer review of "Identification of Characteristic Points in Multivariate Physiological Signals by Sensor Fusion and Multi-Task Deep Networks"

_sensors, 2022, doi:10.3390/s22072684_

Round 1
Reviewer 1 Report
1. The objective (objectives) of one paper must be clearly defined. The "characteristic points in biological signals" must be defined (eventually with example). Also
2. The authors should be very clear when use the terms. "1-lead ECG (EKG) recorders are normally used primarily for basic heart monitoring", so it is a single signal depending on time. "A Multivariate Time Series consist of more than one time-dependent variable and each variable depends not only on its past values but also has some dependency on other variables"
The signals presented by authors should be clearly multivariate signals, because the multivariate analysis is not the same things as multivariate time series.
3. How is looking the actigram as multivariate time series? This signal was mentioned in Abstract and again once in the text. Is is relevant for proposed algorithm in the paper?
4. How is looking the "extracted points" from line 109? How where compared the location of these points found by proposed algorithm with their labeled by an expert?
5. In Fig. 7, it is interesting to see also on y-axis the values 0.4 and probably 0.5.
Reviewer 2 Report
This work uses a recently developed chest-worn sensor which collects ECG, PPG, PCG and accelerometer data and process them offline to determine clean vs. corrupted data using a deep network.
I can't comment on the deep network architecture as it is complicated and requires extensive use and understanding to provide any feedback.
However, I can comment on the following:
1) Data were collected from only 5 subjects. Moreover, all data were cherry picked during night time recordings from these 5 subjects. First, night time data have much better SNR than day time hence, there is bias towards having more clean data. Second, why did the authors use only 2.5 hours of night time recording? How was this decision come about?
2) training, validation and testing procedures are not optimal. Based on the data presented, it appears that data are imbalanced towards having more clean data than motion corrupted data. Please comment on this issue.
3) Suggest using 10-fold or other variant-fold for training and testing procedures. Better yet, leave-one-out subject procedure is the best for testing generalizability of the proposed method. Even better, take independent subjects' data (thereby increasing the number of subjects to 10) and use them as test data to see how well the proposed method is able to the labeling of various class classification of the 4 biosignals.
4) the results presented based on 5 subjects is marginal and the training is largely optimized to these datasets. Hence, there is no confidence that the proposed method will work well for independent datasets (not trained) or especially when data are collected during day time which will be more proned to motion artifacts.
5) abstract: line 8-9, it is not clear what the numbers represent.
6) please provide some kind of rationale for the choice of 2s and 4s of data segment. Why do we care if 2s of data are bad for example. should you perhaps consider 15 or 30 sec of data instead? what is the purpose of defining the signal quality for such a short segment especially when you have many continuous data?
7) what is the computational speed of the proposed DL?
Reviewer 3 Report
I read with interest a very extensive and well-crafted manuscript. Theoretical background is written very well, as well as the following chapters - methodological procedures, results and discussions. I have only a few small recommendations for the whole work, on the formal side:
- The authors state that the study was performed on five men; it would be appropriate to state why this number of participants was determined and also why the survey was conducted only in the male population
- Abbreviations are well explained throughout the text, as the whole manuscript is very extensive, it would be appropriate to re-explain each abbreviation for all graphs / figures.
- Every work should have a conclusion; the conclusion is not a separate chapter in the presented manuscript, although the authors present the conclusions in the last chapter of discussion 4.3; conclusion as a separate chapter should not be missing
- In my opinion, the submitted work fits well into the journal sensor and after minor modifications, including language modifications, I believe that it could be published and that it can be beneficial not only for clinical practice.
Round 2
Reviewer 1 Report
The paper improved and all the requirements are fulfilled.
Reviewer 2 Report
The authors have not adequately addressed my previous comments. The authors have largely avoided answering my questions. Again, my main problem is the fact that the results are based on only 5 subjects. For statistical analysis at minimum, 10 subjects are required. Without validation from at least 10 subjects the results are all speculative and not significant.
